# Cultivation of Protozoa Parasites In Vitro: Growth Potential in Conventional Culture Media versus RPMI-PY Medium

**DOI:** 10.3390/vetsci10040252

**Published:** 2023-03-27

**Authors:** Germano Castelli, Eugenia Oliveri, Viviana Valenza, Susanna Giardina, Flavia Facciponte, Francesco La Russa, Fabrizio Vitale, Federica Bruno

**Affiliations:** Centro di Referenza Nazionale per le Leishmaniosi (C.Re.Na.L.), WOAH Leishmania Reference Laboratory, Istituto Zooprofilattico Sperimentale della Sicilia, Via Gino Marinuzzi 3, 90129 Palermo, Italy

**Keywords:** protozoa parasites, culture media, RPMI-PY medium, *Leishmania* spp., *Trypanosoma cruzi*

## Abstract

**Simple Summary:**

Many different media have been used to grow protozoa parasites, which can be classified into two main categories: biphasic and liquid. All major biphasic media require blood as an essential component for parasite replication: Evans’s modified Tobie and Novy–MacNeal–Nicolle media are the conventional media used for the cultivation of Leishmania and Trypanosoma parasites. These media are time-consuming and often complex, and they are based on relatively expensive reagents and blood components, which complicate immunological and biochemical studies. Our study shows that a new liquid medium, RPMI-PY, can be used for both diagnostic and research purposes since comparable and sometimes better results were found for parasite load compared with the reference media in Leishmania and Trypanosoma. The relative simplicity of RPMI-PY medium, the general availability of its components, and above all, the lack of fresh animal blood requirement renders this medium the best choice for many purposes in protozoa parasites studies. The elimination of fresh blood components is important for economic and ethical aspects, with no need for rabbits’ housing and bleeding, as well as to avoid the complication of the blood components of media in immunological and biochemical studies. Our approach allowed us to clearly evaluate the significant changes in the biological parameters of promastigotes in each medium.

**Abstract:**

The in vitro cultivation of Leishmania and Trypanosoma parasites plays an important role in the diagnosis and treatment of parasitic diseases. Although Evans’s modified Tobie and Novy–MacNeal–Nicolle media, for *Leishmania spp.* and *Trypanosoma cruzi*, respectively, are the two commonly used media for both isolation and maintenance of strains in vitro, their preparation is expensive and laborious and requires fresh rabbit blood from housed animals. The purpose of this study was to evaluate the in vitro growth of both parasites with an alternative monophasic, blood-free, easy, and affordable medium called RPMI-PY, which was previously demonstrated suitable for the in vitro growth of *Leishmania infantum*. The potential growth of different Leishmania species and *Trypanosoma cruzi* was evaluated in traditional culture media versus RPMI-PY medium, and we recorded the protozoa parasites’ morphology via orange acridine–ethidium bromide staining. The results of our study show that RPMI-PY medium can be used for *Trypanosoma cruzi*, *Leishmania amazonensis*, *Leishmania major*, *and Leishmania tropica* species since in all the species except *Leishmania braziliensis*, the exponential growth of the parasite was observed, in many cases higher than conventional media. The staining confirmed not only their growth during the 72 h investigation but also the optimal morphology and viability of the protozoa in the RPMI-PY medium.

## 1. Introduction

Hemoflagellates include parasitic protozoa of medical relevance to humans and other vertebrates. Leishmania and Trypanosoma are the two genera in the group that have a global impact on human health [1]. Both are dixenous genera that infect vertebrates, including humans, through the transmission of different species of blood-feeding (hematophagous) insect vectors [2]. There are about 53 species of Leishmania, and at least 20 of them are infectious to humans [3]. Although distinct Leishmania species morphologically are similar, they mainly involve two clinical forms: cutaneous leishmaniasis (CL) and visceral leishmaniasis (VL) [4]. In CL, parasites infect resident macrophages in the skin, and when the host cell is full of parasites, it bursts, releasing amastigotes that infect neighboring macrophages [5]. In VL, on the other hand, the released amastigotes are spread from the bloodstream and infect cells of the mononuclear phagocytic system of lymph nodes, bone marrow, spleen, etc. [6].

Epidemiologically, visceral leishmaniasis may be of human origin when caused by *Leishmania donovani* or zoonotic when caused by *Leishmania infantum*; in the latter case, dogs are the main hosts. Canine leishmaniasis is found in about 50 of the 88 countries where human leishmaniasis is found, affecting 3 main foci: the Mediterranean Basin, China, and Brazil. Dogs (*Canis familiaris*) are thought to be the primary reservoir of this protozoan and of the zoonotic VL, which can be fatal if left untreated. In canine leishmaniasis, the Leishmania distribution is extensive (liver, lymph glands, spleen, skin, bone marrow, etc.) compared with humans, where the parasite is usually confined to the spleen, bone marrow, and liver. Infection in cats is the same Leishmania genus that affects dogs, and cats from areas endemic to canine leishmaniasis, especially in Mediterranean countries, have been documented to be infected with *L. infantum* [7]. 

In humans, *Trypanosoma cruzi* (*T. cruzi*) induces Chagas disease, a potentially life-threatening disease of the heart and gastrointestinal tract, transmitted by insects of the subfamily Triatominae; the infectious cells are contained in the feces of the insect and can penetrate into the skin of the host [8]. Most cases of acute infection occur in childhood and are usually asymptomatic, although severe myocarditis and meningoencephalitis can occur, and about 30% of people infected with *T. cruzi* will develop the chronic phase of the disease [9,10].

*T. cruzi* shows three morphologically and physiologically distinct primary evolutionary forms during its life cycle, called trypomastigote, epimastigote, and amastigote [11]. Amastigotes and trypomastigotes from the bloodstream are present in the mammalian host, while promastigotes, epimastigotes, and metacyclic trypomastigotes are present in the vector insect [12]. The latter process can be reproduced in vitro using various specific growth conditions.

The diagnosis of these diseases is confirmed with the demonstration of the presence of the parasite in the relevant tissue via the light microscopic examination of the stained sample in in vitro culture, as well as indirectly via a positive serology result and nucleic acid analysis [13,14,15]. The biology of protozoa parasites can be studied in different models, such as insect and mammalian hosts, as well as in in vitro culture. Parasitological diagnosis is always a laboratory diagnosis, and clinical signs and/or clinicopathological abnormalities compatible with the disease lead us to suspect that patients are infected. The confirmation of infection can be obtained using various indirect and direct laboratory diagnostic methods: molecular, serological, and parasitological. Parasitological methods include microscopic examination and culture. The microscopic observation of protozoa parasites in stained bone marrow or lymph node smears is a conclusive diagnosis but requires experience and time. Clinical status and positive immunological/molecular assays may be prognostic, but the “gold standard” diagnostic test is the identification of the parasite through isolation on a specific culture medium [15,16]. The isolation of protozoa parasites from culture is not always successful, and this may be due to several factors, such as the parasite load of the biological sample, the virulence of the strain, and above all, the interaction of the parasites with the ingredients that supplement the culture medium used.

The in vitro culture of parasitic protozoa is an important method because it provides not only information on the parasite but also possibilities for new approaches for the eradication of the parasite and/or containment. It has several fields of application: In addition to diagnostics, it has applicability in research fields, for instance, to study the role of cultures in vaccine development since its continuous use in culture over long periods of time can cause the attenuation of strains; as cell culture in in vivo experimental studies, it provides a large number of parasites for use in experimental animal models; and finally, it is used in studies on the biochemistry, physiology, and metabolism of protozoa and for understanding the ultrastructural organization of the parasite [17].

For the cultivation of protozoa parasites, many different media have been used, which are mainly classified into two categories: biphasic and liquid [18,19,20]. All major biphasic media require blood as an essential component for parasite growth [21] and are used to isolate, maintain, or produce large quantities of protozoa parasites for experimentation [1]. The promastigotes/trypomastigotes forms have been grown in a variety of media at temperatures below 28 °C [22,23], and biphasic media are routinely used for maintenance, producing yields of 10^7^ to 10^8^ cells/mL [24]. Blood is an essential factor of these media, with rabbit blood preferable to other types [24,25].

Evans’s modified Tobie medium (EMTM) and the Novy–MacNeal–Nicolle (NNN) medium are the most conventional media used for the cultivation of *Leishmania* spp. and Trypanosoma parasites [26]. However, several more simple liquid media have been recently developed for the cultivation of Leishmania [24,27,28,29,30,31,32,33]. Castelli et al. (2014) described a new medium called RPMI-PY [26], a monophasic, easy-to-prepare, and inexpensive liquid medium for in vitro *Leishmania infantum* (*L. infantum*) cultivation. The RPMI-PY medium revealed higher efficacy of *L. infantum* growth potential at 24 and 48 h than the traditional EMTM growth medium.

In the present work, an RPMI-PY medium was used to support the continued growth of other hemoflagellate species, such as Leishmania strains belonging to the species *L. major* and *L. tropica*, responsible for distinct clinical manifestations in the Old World, and *L. braziliensis* and *L. amazonensis*, which is present in the New World. In addition, the in vitro growth of *T. cruzi* epimastigotes was evaluated in the new growth medium compared with that in the NNN medium.

## 2. Materials and Methods

### 2.1. Media Preparation

#### 2.1.1. Evans’s Modified Tobie Medium

The biphasic EMTM was composed of two parts: a solid phase and a liquid phase. The “solid phase” consisted of 3 g beef extract, 5 g bacteriological peptone, 20 g agar, 8 g NaCl, and distilled water until 1000 mL. The resulting solution was homogenized and sterilized in an autoclave, after which defibrinated rabbit blood was also added (15 mL/100 mL). The “liquid phase” consisted of 0.2 g KCl; 0.03 g Na_2_HPO_4_ 12H_2_O, 0.03 g KH_2_PO_4_; 0.07 g CaCl_2_; 0.05 g MgSO_4_ 3 7H_2_O; 0.05 g MgCl_2_ 3 6 H2O; 4 g NaCl; and distilled water until 500 mL. The mixture was homogenized and autoclaved. The liquid phase was finally supplemented with L-Proline (0.1 g in 100 mL), 10% fetal bovine serum (SFB) decomplemented at 56 °C for 30 min, 5% human urine, a 1% phenol red solution, a 2% antibiotic solution (250 ug/mL gentamicin, and 500 ug/mL 5-fluorocytosine). The pH was adjusted to 7,18, and the liquid phase was also filter-sterilized. EMTM was finally completed by combining 4 mL of the solid-phase medium with 1.5 mL of the liquid-phase medium dispensed into 25 cm^2^ flasks [26,34,35].

#### 2.1.2. Novy–MacNeal–Nicolle Medium

The NNN medium was a biphasic medium composed of a blood agar base (0.3 g Meat extract; 0.5 g peptone; 0.8 g; sodium chloride; 1.5 g agar; and 100 mL of distilled water, with the pH adjusted to 7.3 ± 0.2), autoclaved and enriched with 10% of sterile defibrinated rabbit; and a “liquid-phase” component (0.02 g potassium chloride; 0.8 g sodium chloride; 0.02 g calcium chloride; 0.03 g Anhydrous potassium dihydrogen phosphate; and 0.25 g dextrose; final pH 7.0 ± 0.2) autoclaved. NNN was completed by combining 4 mL of the solid-phase medium with 2 mL of the liquid-phase medium dispensed into 25 cm^2^ flasks [36].

#### 2.1.3. RPMI-PY Medium

RPMI-PY medium was composed as follows: RPMI 1640 (Merk^®^ RPMI Media 1640) integrated with the same volume of P-Y medium, with 10% SFB and 1% glutamine, 250 µg/mL gentamicin, and 500 µg/mL 5-fluorocytosine. Following Limoncu et al. (1997) [24], the PY culture medium consisted of 1 g peptone; 0.8 g NaCl; 0.75 g Na_2_HPO_4_; 0.25 g yeast extract; and distilled water to 100 mL. The medium was then passed through a PVDF filter with a pore diameter of 0.22 µm. Then, 10 mm were distributed in 25 cm^2^ flasks.

### 2.2. Protozoa Parasites

Four different hemoflagellate parasites, namely *L. braziliensis* (MHOM/BR/75/M2904), *L. major* (MHOM/SU/73/5ASKH), *L. tropica* (MHOM/SU/74/K27), *L. amazonensis* (IFLA/Br/67/PH8), and *Trypanosoma cruzi*, were used in the experiments and were cultured at 25 °C, pH 7,18 in EMTM for Leishmania strains and NNN for Trypanosoma, respectively. Parasite growth was qualitatively and quantitatively monitored via daily microscopic observations for cell mobility and viability. The logarithmic phase was adjusted to a final concentration of 4 × 10^6^/mL of parasites, grown in 25 cm^2^ cell culture flasks, and incubated at 25 °C. The cultures were monitored by measuring the growth of parasites over a 72 h incubation period. Parasites were counted using a Bürker counting chamber under standard light microscopy (20× objective) after 24 h, 48 h, and 72 h.

### 2.3. Morphological Analysis

The protozoa parasites were stained with a mixture of ethidium bromide (100 mg/mL) and acridine orange (100 mg/mL) [37,38]. Promastigotes/epimastigotes (4 × 10^6^/mL) were suspended in 25 cm^2^ flasks containing 10 mL of RPMI-PY and in 25 cm^2^ containing conventional media. After 24, 48, and 72 h, the cells (1 × 10^6^) were centrifuged, and the pellet was resuspended in 25 mL of the stained mixture and examined with a 40× objective using a Leica DM 4000B fluorescence microscope (Leica, Heerbrugg, Switzerland) to analyze the morphological changes that the parasites assumed in the culture media [37,38].

### 2.4. Statistical Analysis 

Statistical analyses were performed using GraphPad Prism 9.0 (GraphPad Software Inc., San Diego, CA, USA). All data were obtained from three independent experiments, and the results are shown as mean ± standard deviation or as a representative experiment. The assumption of the normality of the distribution of data groups was tested with a Shapiro–Wilk test. All the data groups were normally distributed; therefore, a two-way ANOVA test was carried out. A *p*-value  <  0.05 was considered statistically significant.

## 3. Results

The objective of the study was to demonstrate the growth potential of protozoa parasites belonging to the genera Leishmania and Trypanosoma in blood-free culture media. We started by evaluating the biological characteristics and growth behavior of *Leishmania* spp. and *Trypanosoma cruzi* strains in the new RPMI-PY growth medium, compared over time with conventional media.

Figure 1 shows the growth viability of *L. amazonensis*, *L. braziliensis*, *L. major*, and *L. tropica* at 24 h, 48 h, and 72 h in EMTM and RPMI-PY. *L. amazonensis* showed a higher number of parasites at 24 h in EMTM (2 × 10^7^/mL), while significant growth was observed in RPMI-PY at 48 and 72 h. In particular, at 48 h, *L. amazonensis* in RPMI-PY showed a modest increase in the number of parasites (6.2 × 10^7^/mL) compared with the growth of parasites in the conventional medium (3.3 × 10^7^/mL); at 72 h, however, an increase in parasite growth of one log (3.9 × 10^8^/mL) was observed in RPMI-PY compared with EMTM medium (5.6 × 10^7^/mL) (Figure 1a). *L*. *braziliensis* showed a different growth pattern in RPMI-PY versus EMTM, compared with the other Leishmania species analyzed, with exponential growth only observed in conventional culture medium (EMTM, 24 h: 5 × 10^7^/mL, 48 h: 2.5 × 10^7^/mL, and 72 h: 3.9 × 10^8^/mL; RPMI-PY, 24 h: 2.3 × 10^6^/mL, 48 h: 9.8 × 10^6^/mL, and 72 h: 2.9 × 10^7^/mL) (Figure 1b). The growth curves of *L. major* in the two different growth media are shown in Figure 1c. In both media, parasites showed exponential growth over time, although the number of parasites was higher in the new liquid medium (24 h: 3 × 10^7^/mL; 48 h: 8.3 × 10^7^/mL; and 72 h: 3.8 × 10^8^/mL) than in EMTM (24 h: 8 × 10^5^/mL; 48 h: 1.6 × 10^7^/mL; and 72 h: 1.4 × 10^8^/mL). *L. tropica* exhibited different growth curves between the biphasic and liquid media; in the EMTM, a growth peak was observed to be reached at 48 h (5.3 × 10^8^/mL) decreasing at 72 h (3.3 × 10^8^/mL), whereas in RPMI-PY, exponential growth was observed over time (24 h: 3.6 × 10^7^/mL; 48 h: 3.3 × 10^8^/mL; and 72 h: 8.7 × 10^8^/mL) (Figure 1d).

*Trypanosoma cruzi* showed exponential growth over time, in both culture media, with no statistically significant differences between the NNN biphasic medium and the RPMI-PY liquid medium. Trypanosoma in NNN medium was counted as 2.4 × 10^7^/mL at 24 h, 4.2 × 10^7^/mL at 48 h, and 8.2 × 10^7^/mL at 72 h; in comparison, in RPMI-PY, it was found to be 2.1 × 10^7^/mL at 24 h, 3.7 × 10^7^/mL at 48 h, and 7.2 × 10^7^/mL at 72 h (Figure 2).

To gather the biological information more comprehensively, we also recorded the dominant morphology of the parasites in RPMI-PY at specific times (Figure 3). In Leishmania, acridine orange and ethidium bromide (OA/BE) staining was primarily used to distinguish between live and damaged or dead cells [39]. On the first day of culture, after labeling with OA/BE, the cells showed green fluorescence and classical morphology being slender, flagellated, and elongated in both RPMI-PY and EMTM medium. Within 24 h in the cultures of *L. amazonensis* and *L. braziliensis*, more cells were observed in EMTM than in RPMI-PY, whereas more cells were observed in RPMI-PY in the cultures of *L. tropica* and *L. major*. In addition, fluorescence microscopy with OA/BE double staining revealed that cell death was observed in the *L. major* group in EMTM compared with the RPMI-PY culture. In contrast, in the RPMI-PY cultures of *L. braziliensis*, cells with greater chromosome condensation were observed than in EMTM cultures [39,40,41]. At 72 h, however, there was an increase in the number of green, viable promastigotes in all Leishmania strains in RPMI-PY compared with EMTM, with the exception of *L. braziliensis* [27,28,29,30,31,32]. RPMI-PY determined viable and uniform cells in cultures of *L. major*, *L. amazonensis*, and *L. tropica.* Almost all promastigotes had long, thin, and progressively tapered bodies, with flagella one and a half to two times as long as the body, while *L. braziliensis* in the RMPI-PY medium had consistently smaller bodies. In *L. tropica*, the cytoplasmic membrane appeared somehow broken, and the nucleus appeared red in both media at 72 h.

The morphological analysis revealed no changes in the form and size of Trypanosomes in the RPMI-PY and NNN media analyzed at 24 and 72 h (Figure 4).

## 4. Discussion

In this study, we described and analyzed the potential growth of parasites of the genera Trypanosoma and Leishmania, in a blood-free growth medium, RPMI-PY medium, by evaluating the biological characteristics and growth behavior of the parasites over 72 h and comparing it with conventional EMTM and NNN media used for Leishmania and Trypanosoma, respectively. RPMI-PY has a relatively simple formulation consisting of common, available, and inexpensive ingredients, which, in terms of time and parasite load, could provide an alternative to conventional media enriched with rabbit blood. After the interesting growth results of *L. infantum* species in the RPMI-PY medium investigated by Castelli et al. [26], our study confirmed the possibility of using this liquid medium for other Leishmania species: *L. amazonensis* (Figure 1a), *L. major* (Figure 1c), and *L. tropica* (Figure 1d), showing a satisfactory growth rate in 48 h and especially in 72 h, compared with conventional EMTM medium, except *L. braziliensis* (Figure 1b). The exception of *L. braziliensis* may be due to its different nutritional requirements for some nutrients, particularly folate and pterin, which are two essential nutrients for the parasite [42]. Folic acid has been implicated in the synthesis of thymidine and methionine and the interconversion of serine into glycine by Leishmania [43]. In *L. braziliensis*, this is particularly important, as it has been reported that this species requires higher concentrations of folic acid than other Leishmania species [42,43]. The results also showed that RPMI-PY could also be used satisfactorily for the cultivation of *Trypanosoma cruzi*. In fact, the parasite showed exponential growth at 72 h as in the NNN reference medium. The presented results show that metacyclic trypomastigotes of *infectious T. cruzi* can be obtained in large quantities under chemically defined conditions. Consequently, this medium can provide an important tool for studying the immunoprophylaxis of Chagas disease.

Parasite viability, analyzed by double staining with acridine orange and ethidium bromide, confirmed the results of cell counts; in RPMI-PY, the cells appeared green, with a classic slender, elongated, and flagellated morphology, at 24 h as well as 48 and 72 h. Thus, the staining confirmed not only the optimal morphology and viability of the protozoa but also their growth during the 72 h investigation compared with conventional media.

Cultivation techniques are very important for deeper analysis of many pathogens including protozoa parasites [44], and success in establishing in vitro cultures of parasites makes it possible to dynamically study their metabolism, physiology, and behavior and to analyze the nature of the antigenic molecules present in their excretory and secretory products. In addition, in the field of molecular biology, they enable the better characterization of strains and the potential development of defective, avirulent strains to be used as vaccines to protect humans and animals from diseases. Conventional classifications of Leishmania and Trypanosoma have been based on the clinical manifestations caused by different strains or species [24], and within species, there are variations in the ability to grow in particular media, with some strains more readily lending themselves to in vitro cultivation. Using different materials in biphasic and liquid media, numerous culture media [19,24,29,33,45] have been used for the culture of protozoa parasites and compared with each other [24,45]. Many culture media are used to cultivate protozoa parasites: The NNN medium with agar and defibrinated rabbit blood is the preferred culture medium for protozoa parasites, and together with EMTM medium with beef, agar, and peptone, are two widely used biphasic media. Although biphasic media allow considerable parasite growth, they have the disadvantages of being complex, time-consuming, and expensive [35]. In addition, the use of fresh animal blood in culture is problematic for the management of animal houses with all the related ethical aspects. Moreover, the blood components of the cultural media can interfere with immunological and biochemical studies. Considering that RPMI-PY is a much easier medium to prepare, with a relatively simple formulation consisting of common, available, and inexpensive ingredients for in vitro maintenance and mass cultivation of different species of Leishmania and Trypanosoma, it is already a great advantage that can be useful in laboratory routine.

## 5. Conclusions

This study shows that RPMI-PY can be used for both diagnostic and research purposes since comparable and sometimes better results for parasite load were found compared with the reference media, namely NNN for *T. cruzi* and EMTM for *Leishmania* spp. The relative simplicity of RPMI-PY formulations, the general availability of its components, and above all, the lack of fresh animal blood requirement renders this media the best choice for many purposes in protozoa parasites studies. The elimination of fresh blood components is important for both economic and ethical aspects (no need for rabbits’ housing and bleeding), as well as to avoid the complication of the media blood components in immunological and biochemical studies. Our approach enabled us to clearly assess significant alterations in relevant promastigotes’ biological parameters in each medium. RPMI-PY has the biological characteristics required for becoming a medium of choice in the maintenance of these parasites.

## Figures and Tables

**Figure 1 vetsci-10-00252-f001:**
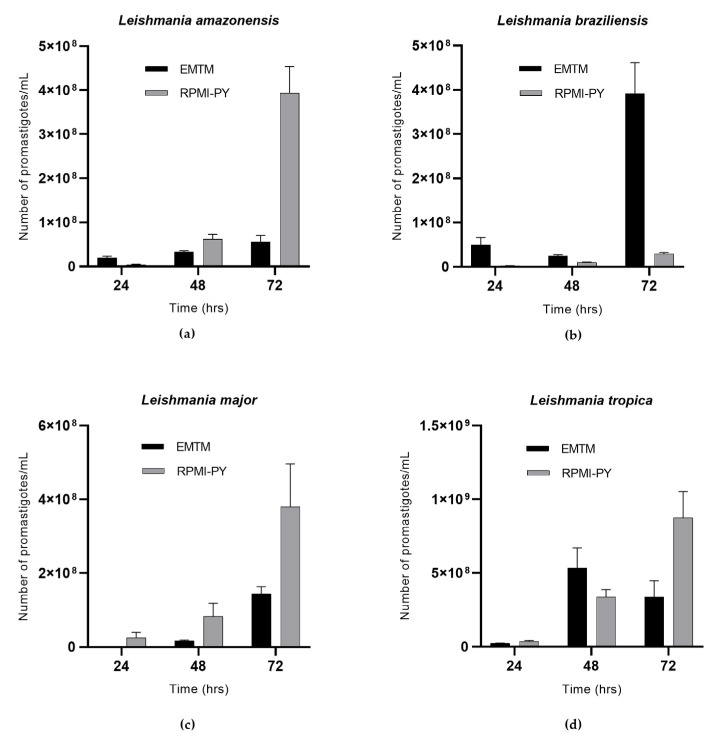
Growth curves of Leishmania spp. in RPMI-PY and EMTM media with 4 × 10^6^/mL inoculation in 25 cm^2^ flasks at 26 °C observed over a 3 days period under a light microscope: (**a**) *Leishmania amazonensis*; (**b)**
*Leishmania braziliensis*; (**c**) *Leishmania major*; (**d**) *Leishmania tropica*. Bars show the mean ± SD. The significant difference in parasite growth rate for different media was determined using two-way analysis of variance (ANOVA) analysis and demonstrated as all values showing a *p* < 0.05.

**Figure 2 vetsci-10-00252-f002:**
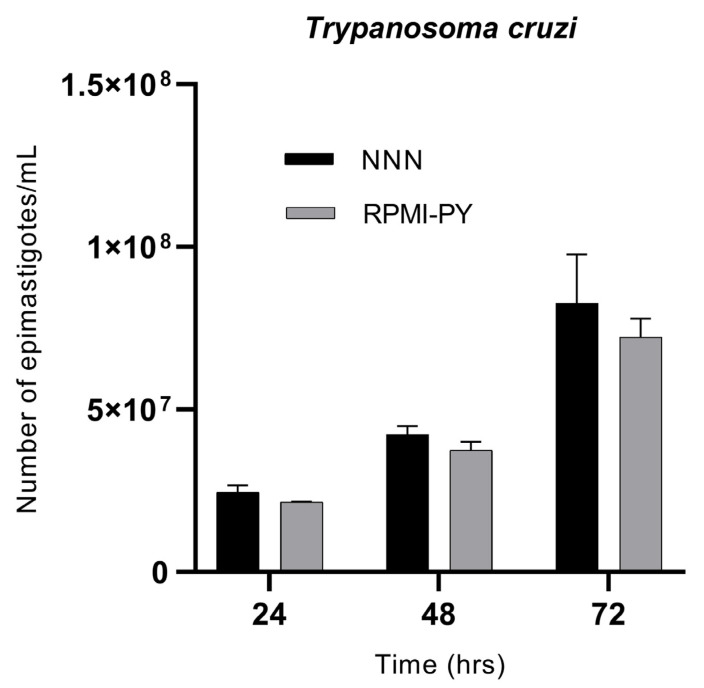
Growth kinetics of *Trypanosoma cruzi* parasites/mL in NNN and RPMI-PY media at 24 h, 48 h, and 72 h. Values represent the mean of triplicates. Error bars indicate standard deviations. The significant difference in parasite growth rate for different media was determined via two-way analysis of variance (ANOVA) analysis and demonstrated as all values showing a *p* < 0.05.

**Figure 3 vetsci-10-00252-f003:**
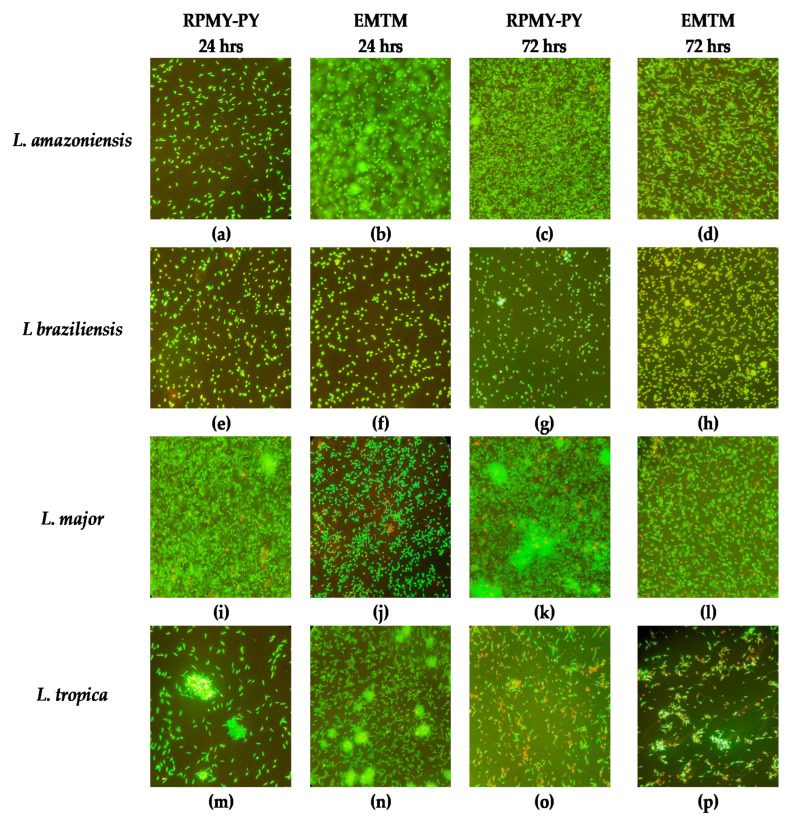
Morphologic changes observed in Leishmania stained with ethidium bromide and acridine orange. Live cells were identified by uptake of acridine orange (green fluorescence) and exclusion of ethidium bromide (red fluorescence): (**a**) *Leishmania amazonensis* in RPMI-PY for 24 h; (**b**) *Leishmania amazonensis* in EMTM for 24 h; (**c**) *Leishmania amazonensis* in RPMI-PY for 72 h; (**d**) *Leishmania amazonensis* in EMTM for 72 h; (**e**) *Leishmania braziliensis* in RPMI-PY for 24 h; (**f**) *Leishmania braziliensis* in EMTM for 24 h; (**g**) *Leishmania braziliensis* in RPMI-PY for 72 h; (**h**) *Leishmania braziliensis* in EMTM for 72 h; (**i**) *Leishmania major* in RPMI-PY for 24 h; (**j**) *Leishmania major* in EMTM for 24 h; (**k**) *Leishmania major* in RPMI-PY for 72 h; (**l**) *Leishmania major* in EMTM for 72 h; (**m**) *Leishmania tropica* in RPMI-PY for 24 h; (**n**) *Leishmania tropica* in EMTM for 24 h; (**o**) *Leishmania tropica* in RPMI-PY for 72 h; (**p**) *Leishmania tropica* in EMTM for 72 h.

**Figure 4 vetsci-10-00252-f004:**
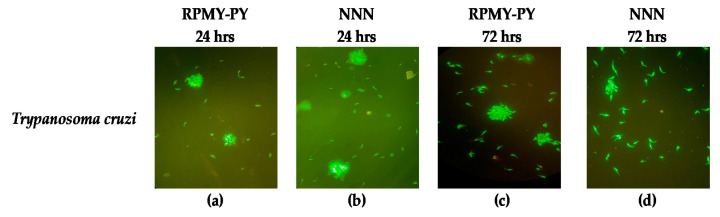
Morphologic changes observed in *Trypanosoma cruzi* stained with ethidium bromide and acridine orange. Live cells were identified via the uptake of acridine orange (green fluorescence) and exclusion of ethidium bromide (red fluorescence): (**a**) *Trypanosoma cruzi* in RPMI-PY for 24 h; (**b**) *Trypanosoma cruzi* in NNN for 24 h; (**c**) *Trypanosoma cruzi* in RPMI-PY for 72 h; (**d**) *Trypanosoma cruzi* in NNN for 72 h.

## Data Availability

Not applicable.

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
