# Peer review of "Cultivation of Protozoa Parasites In Vitro: Growth Potential in Conventional Culture Media versus RPMI-PY Medium"

_vetsci, 2023, doi:10.3390/vetsci10040252_

Round 1

Reviewer 1 Report

Title: Cultivation of hemoflagellates parasites in vitro: growth potential in conventional culture media versus RPMI-PY medium

 Overview: Castelli and colleagues present an article whose aim of the work was to test a new culture medium for use in protozoa parasites.

The idea of ​​the manuscript is to suggest a cheaper and simpler culture medium to be made, which would help many research works. However, some concepts need to be revised and changes should be made to the text.

Therefore, in the present form, I do not consider the manuscript for publication. I recommend a full revision of the paper.

 1) I recommend changing the title of the article and removing the word hemoflagellates. A suggestion would be “Cultivation of protozoa parasites...”.

2) The word hemoflagellates is generally used for forms of the parasite that circulate in the blood. Few authors consider leishmania as a hemoparasite because its forms do not circulate freely in the blood. This topic is a bit controversial. I recommend not using this term. Especially because the forms used in the experiments were Leishmania promastigotes and Trypanosoma epimastigotes, which are forms of the parasite found in the vector and in the culture medium and are not found circulating in the blood. Then the text must be revised.

3) In the abstract the end of the text should be revised and changed. It concludes that the proposed medium can be a good alternative for Leishmania spp but actually did not work for all species tested. So I suggest you put the species that worked (line 43).

4) Line 89 – the name L. amazonensis is spelled wrong.

5) Lines 126 and 138 – delete the name hemoflagellate in the text.

6) Figure 1 legend – its not necessary repeat the “growth curve of...”  a) L. amazonensis; b) L. braziliensis....

7) Figure 3 legend -  the name L. amazonensis and L. braziliensis are spelled wrong (lines 226-229).

Author Response

Dear,

Thank you for giving me the opportunity to submit a revised draft of my manuscript titled "Cultivation of protozoa parasites in vitro: growth potential in conventional culture media versus RPMI-PY medium”. We appreciate the time and effort that you have dedicated to providing your valuable feedback on manuscript. We have been able to incorporate changes to reflect most of your suggestions.

  • We change the title in: “Cultivation of protozoa parasites in vitro: growth potential in conventional culture media versus RPMI-PY medium”
  • We have also replaced the word “hemoflagellates” with “protozoan parasites” throughout the entire rest of the manuscript
  • We have corrected the wrong name of amazonensis and L. brazilensis in the text and in Figure 3 legend.
  • In Figure 1 legend we removed “growth curve of…” as suggested.
  • We have modified the abstract including “The results of our study showed that RPMI-PY medium can be used for Trypanosoma cruzi, Leishmania amazonensis, Leishmania major, Leishmania tropica species since in all species except Leishmania braziliensis, exponential growth of the parasite was observed, in many cases higher than conventional media. The staining confirmed not only their growth that occurred over the 72 hour investigation, but also the optimal morphology and viability of the protozoa in RPMI-PY medium.” Instead of “Different parasite species showed parasite growth rate in RPMI-PY was significantly equal or higher than Tobie's modified Evans medium and Novy-MacNeal-Nicolle medium at 24, 48 and 72 hours. These results suggest that this bloodless liquid medium RPMI-PY can be used as a valid alternative to the more difficult to prepare and expensive media for isolating and culturing Leishmania spp. and Trypanosoma cruzi strains.”

We look forward to hearing from you in due time regarding our submission and to respond to any

further questions and comments you may have.

Sincerely,

Reviewer 2 Report

This manuscript contains some considerable data but first needs to be revised with respect to several important issues. 

Abstract - requires a  correction (too general). The Abstract does not summarize the paper’s discussions.

Statistical analyses used in the experiment need to be more detailed. Statistical analysis: Did the authors assessed if the data was normal distributed before using test? If so, please state how this assessment was made.

This section is vague and does not cover the whole aspects necessary for the interpretation of the work undertaken. It may be rather wanders, making it difficult to understand the overall importance and the essential meaning of the work. So the authors need to summarize in the first paragraph of discussion fully documented test results and on this basis - what the general findings were, and then give the important aspects bearing discussion, which alluded directly to the essential statements and conclusions arising from the work.

The premises of the research are very interesting, but I feel unsatisfied after reading the manuscript. The presentation of results and their interpretation are rather weak.  The paper can be published after necessary additions and correction of some details. The discussion should be extended. 

Author Response

Dear,

Thank you for giving me the opportunity to submit a revised draft of my manuscript titled Cultivation of protozoa parasites in vitro: growth potential in conventional culture media versus RPMI-PY medium”. We appreciate the time and effort that you have dedicated to providing your valuable feedback on manuscript. We have been able to incorporate changes to reflect most of your suggestions.

  • We have modified the abstract including “The results of our study showed that RPMI-PY medium can be used for Trypanosoma cruzi, Leishmania amazonensis, Leishmania major, Leishmania tropica species since in all species except Leishmania braziliensis, exponential growth of the parasite was observed, in many cases higher than conventional media. The staining confirmed not only their growth that occurred over the 72 hour investigation, but also the optimal morphology and viability of the protozoa in RPMI-PY medium.” Instead of “Different parasite species showed parasite growth rate in RPMI-PY was significantly equal or higher than Tobie's modified Evans medium and Novy-MacNeal-Nicolle medium at 24, 48 and 72 hours. These results suggest that this bloodless liquid medium RPMI-PY can be used as a valid alternative to the more difficult to prepare and expensive media for isolating and culturing Leishmania spp. and Trypanosoma cruzi strains.”
  • Lines 183-192: we have changed statistical analyses with “Statistical analyses were performed using the GraphPad Prism 9.0 (GraphPad Software Inc., CA, USA). All data were obtained from three independent experiments and the results are shown as mean ± standard deviation or as a representative experiment. The assumption of normality of distribution of data groups was tested by a shapiro- wilk test. All the data groups were normally distributed therefore, a two-way ANOVA test was carried out to. A P-value < 0.05 was considered statistically significant.”
  • We have modified the discussion section: “This study describes and analyzes the parasites potential growth of the genera Trypanosoma and Leishmania, in a blood-free growth medium, RPMI-PY medium, by evaluating the biological characteristics and growth behavior of the parasites over 72 hours, comparing it with conventional EMTM and NNN media used for Leishmania and Trypanosoma, respectively. The RPMI-PY medium has a relatively simple formulation consisting of common, available and inexpensive ingredients, which could provide in terms of time and parasite load an alternative to conventional media enriched with rabbit blood. After the interesting growth results of infantum species in the RPMI-PY medium investigated by Castelli et al. [24], our study confirmed the possibility of using this liquid medium for other Leishmania species: L. amazonensis (Figure 1 a), L. major (Figure 1c) and L. tropica (Figure 1 d), showing a growth rate in 48 hours and especially in 72 hrs satisfactory, compared with conventional EMTM medium, except L. braziliensis (Figure 1b). The exception of L. braziliensis may be due to its different nutritional requirements for some nutrients particularly folate and pterin which are two essential nutrients for the parasite [40]. Folic acid has been implicated in the synthesis of thymidine and methionine and the interconversion of serine into glycine by Leishmania [41]. In L. braziliensis, this is particularly important as it has been reported that this species requires higher concentrations of folic acid compared to other Leishmania [40,41]. The results also showed that RPMI-PY could also be used satisfactorily for the cultivation of Trypanosoma cruzi. In fact, the parasite showed exponential growth at 72 hrs as in the NNN reference medium. The parasites viability, analyzed by double staining with acridine orange and ethidium bromide, confirmed the results of cell counts, in RPMI-PY the cells appeared green, with the classic slender, elongated and flagellated morphology, at 24 hrs as well as 48 and 72 hrs. Thus,the staining not only confirmed the confirmed not only the optimal morphology and viability of the protozoa, but also their growth that occurred in the 72 hrs investigation compared with conventional media. Cultivation techniques are very important for deeper analysis of many pathogens including protozoa parasites [19] and success in establishing in vitro cultures of parasites makes it possible to study their metabolism dynamically, physiology, behavior and to analyze the nature of the antigenic molecules present in their excretory and secretory products. In addition, it enables molecular biology studies for a better characterization of strain and the potentials development of defective, avirulent strains to be used as vaccines to protect humans and animals from the diseases. Conventional classifications of Leishmania and Trypanosoma have been based on the clinical manifestations caused by different strains or species [22], and within species there are variations in the ability to grow in particular media, with some strains lending themselves more readily to in vitro cultivation. Numerous culture media [16,22,27,31,42] have been used for the culture of protozoa parasites, using different materials in biphasic and liquid media, and have been compared to each other [22,43]. Many culture media are used to cultivate protozoa parasites: NNN medium with agar and defibrinated rabbit blood is the preferred culture medium for protozoa parasites, together with EMTM medium with beef, agar and peptone, are the widely used biphasic media. Although biphasic media allow considerable parasite growth, these have the disadvantages of being complex, time-consuming, and expensive [35]. In addition, the use of fresh animal blood in culture are problematic for the management of animal houses with all related ethical aspects. Moreover, the blood components of the cultural media can interfere with immunological and biochemical studies. Considering that RPMI-PY medium is a much easier medium to prepare, with a relatively simple formulation consisting of common, available and inexpensive ingredients for in vitro maintenance and mass cultivation of different species of Leishmania and Trypanosoma, it is already a great advantage that can be useful in laboratory routine.”

We look forward to hearing from you in due time regarding our submission and to respond to any

further questions and comments you may have.

Sincerely,

Round 2

Reviewer 1 Report

I accepted the corrections

Reviewer 2 Report

Accept in present form